# Rapid diagnosis of periprosthetic joint infection from synovial fluid in blood culture bottles by direct matrix-assisted laser desorption ionization time-of-flight mass spectrometry

**Feng-Chih Kuo[1,2], Chun-Chih Chien[3], Mel S. Lee[1,2], Jun-Wen Wang[1,2], Po-Chun Lin[1,2], Chen-Hsiang Lee[2,4]** *

**1** Department of Orthopaedic Surgery, Kaohsiung Chang Gung Memorial Hospital, Kaohsiung, Taiwan,
**2** College of Medicine, Chang Gung University, Kaohsiung, Taiwan, **3** Laboratory Medicine, Kaohsiung Chang Gung Memorial Hospital, Kaohsiung, Taiwan, **4** Division of Infectious Diseases, Department of Internal Medicine, Kaohsiung Chang Gung Memorial Hospital, Kaohsiung, Taiwan

* lee900@cgmh.org.tw

**Data Availability Statement:** All relevant data are within the manuscript and its Supporting Information files.

## Abstract

### Background

The aim of this prospective study was to use direct matrix-assisted laser desorption ionization time-of-flight mass spectrometry (MALDI-TOF MS) to rapidly diagnose periprosthetic joint infections (PJIs).

### Method

Synovial fluid was taken from 77 patients (80 joints, 41 hips and 39 knees) who met the International Consensus Meeting criteria for PJI, and inoculated into blood culture bottles (BCBs) and onto conventional swabs. Positive blood cultures were analyzed using either direct or routine MALDI-TOF MS. Pathogen identification and the time to identification was recorded. Differences between groups were analyzed using the Kruskal-Wallis test and Bonferroni's post-hoc test.

### Results

Direct and routine MALDI-TOF MS both detected 64 positive results (80%), compared to 47 (59%) by conventional swabs (p = 0.002). Direct MALDI-TOF MS identified 85.3% of the gram-positive organisms and 92.3% of the gram-negative organisms. No fungi were identified by direct MALDI-TOF MS. In 17 BCBs that were flagged positive, identification by direct MALDI-TOF MS failed. Among the positive results in the direct MALDI-TOF MS group, *Staphylococcus aureus* accounted for 47%, followed by *Staphylococcus epidermidis* (17%), *Escherichia coli* (9%) and *Klebsiella pneumoniae* (9%). The median time to microorganism identification was significantly shorter with direct MALDI-TOF MS (12.7 h, IQR: 8.9–19.6 h) than with routine MALDI-TOF MS (39.5 h, IQR: 22.8–46.0 h) or swabs (44.4 h, IQR: 27.2–

**Funding:** The Chang Gung Medical Research Program from Chang Gung Medical Foundation provided support for this study in the form of a grant awarded to FCK (CMRPG8F1402). The funders had no role in study design, data collection and analysis, decision to publish, or preparation of the manuscript.

**Competing interests:** The authors have declared that no competing interests exist.

72.6 h) ($p < 0.0001$). In pairwise comparisons, there were significant differences in the time of microorganism identification between direct MALDI-TOF MS and routine MALDI-TOF MS ($p < 0.0001$) or swab culture ($p < 0.0001$). There was no significant difference between routine MALDI-TOF MS and swab culture ($p = 0.0268$).

## Conclusion

Compared with current laboratory practice, direct MALDI-TOF MS shortened the time to microorganism identification and had superior results compared to conventional swabs, except for fungi. Further studies should investigate whether the earlier administration of appropriate antimicrobial agents can improve the treatment outcomes of PJIs.

## Introduction

Prosthetic joint infections (PJIs) remain one of the most catastrophic complications after total joint replacement [1]. In order to improve the prognosis and provide effective organism-specific antibiotic treatment, the causative microorganism must be identified accurately and rapidly.

Conventional cultures of synovial fluid (SF) specimens require an incubation period of 14 days due to slower growing or fastidious organisms [2]. SF samples in blood culture bottles (BCBs) have been shown to enable quicker identification of pathogens in cases of lower extremity PJIs [3]. Previous studies have suggested that inoculating SF into BCBs can increase the sensitivity and specificity for the diagnosis of PJIs with an incubation period of 5 days when using the BACTEC 9240 system (BD Diagnostic Systems, BD Corporation, NJ, USA) [4]. In addition, when combined with an automated blood culture system (BD BACTEC$^{TM}$), the detection of most organisms has been shown to be possible within 3 days using BCBs [5]. Furthermore, another study showed that the time to microorganism detection was shortened to within a median of 21 and 23 hours by culturing periprosthetic tissue specimens in aerobic and anaerobic BCBs, respectively [6].

Recently, matrix-assisted laser desorption ionization time-of-flight mass spectrometry (MALDI-TOF MS) has been used to rapidly identify bacteria in positive BCBs. The major advantage of MALDI-TOF technology in identifying bacteria is the time to obtaining results, which has been reported to be reduced from 24–48 hours to less than an hour when performing routine identification of bacterial colonies grown on defined agar [7, 8]. The most time-intensive process when using routine MALDI-TOF MS is the time required for the subculture of specimens before identification. To overcome this issue, a new technique has been developed to rapidly identify bacteria from positive blood cultures known as direct MALDI-TOF MS, which has a turnaround time that can be shortened to less than 24 hours [9, 10]. Theoretically, direct MALDI-TOF MS can decrease the time to preliminary strain identification, and this rapid microbial identification method has been used in many infection fields, including bloodstream infections [8, 11, 12] meningitis [13, 14] and urinary traction infections [15, 16]. Direct MALDI-TOF MS has been shown to have high accuracy for gram-negative organisms but relatively low accuracy for gram-positive bacteria [17]. Although gram-positive bacteria are the most common organisms that cause PJIs [18, 19], whether direct MALDI-TOF MS of SF cultured in BCBs can be used to rapidly and accurately diagnose PJIs has yet to be elucidated.

The purpose of this study was to evaluate whether: 1) pathogen identification using direct MALDI-TOF MS has similar results to routine MALDI-TOF MS and conventional cultures,

and 2) whether direct MALDI-TOF MS can shorten the time to preliminary strain identification compared with routine MALDI-TOF MS and conventional cultures in patients with PJIs.

## Material and methods

### Study design

The Institutional Review Board of Chang Gung Medical Foundation approved this study (No. 201601286B0C502). Patients who had a high probability of infection based on the International Consensus Meeting (ICM) criteria for the diagnosis of PJI [20] and who were scheduled for debridement only or debridement with implant removal were prospectively enrolled in this study from December 2016 to May 2019. Acute PJI was defined as symptom duration of $< 4$ weeks, and chronic PJI was defined as symptom duration of $\geq 4$ weeks. Patients who did not meet the Musculoskeletal Infection Society (MSIS) criteria, who underwent aseptic revision, or who had an insufficient amount of SF for analysis were excluded from the study. The study was registered in the public ClinicalTrials.gov registry (Identifier: NCT03717090). Informed consent was obtained from all patients.

### Samples

Synovial joint fluid was sampled prior to arthrotomy in the surgical theater. Aspirates were collected using an aseptic technique with an 18-Fr sterile syringe, and a minimum of 14 mL was collected from each patient. Each sample was partitioned between two sets of aerobic and anaerobic BCBs (at least 2.5 mL per bottle), conventional swabs (2 mL), and for synovial white blood cell (WBC) count and neutrophil percentage (PMN %) (2 mL). In instances where there was insufficient fluid for analysis using all three modalities (less than 14 mL), the patients were excluded from the current investigation. Samples were delivered to the clinical microbiology laboratory within a 2-hour period. The diagnostic workflow is shown in Fig 1.

### Conventional swab culture

For conventional cultures, a few drops of well-mixed SF were inoculated onto blood agar plates (BAP)/eosin methylene blue (EMB) agar plates and chocolate agar plates. The plates were incubated at 35˚C in an aerobic atmosphere with 5% carbon dioxide ($CO_2$) for 18–24 hours.

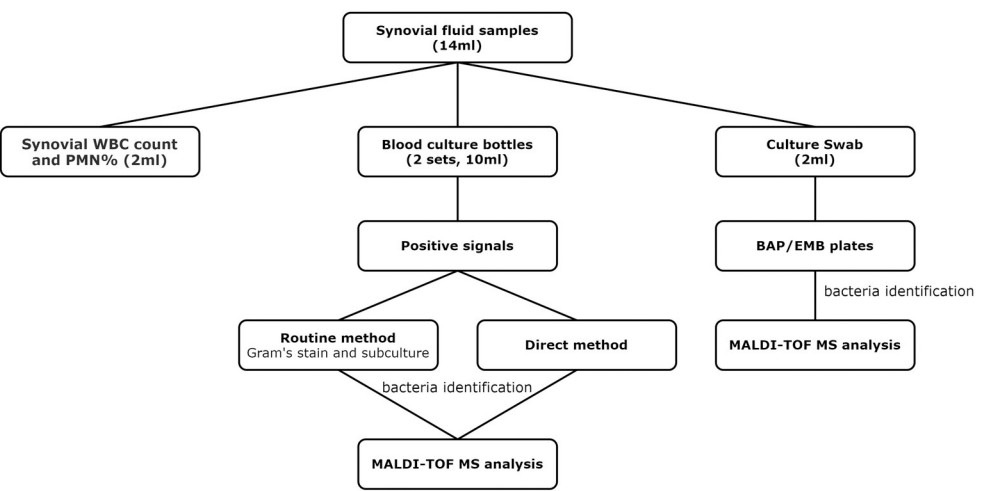

**Fig 1. Schematic representation of the diagnostic workflow.**

When bacterial growth was observed, the colonies on the agar plate were identified using routine MALDI-TOF MS (Bruker Daltonik Bremen, Germany) as described below.

## Sample incubation in an automated blood culture system

Synovial fluid specimens were inoculated into BD BACTEC Plus Aerobic BCBs (Becton, Dickinson and Company Sparks, MD, USA) and incubated in an automated blood culture system (BACTEC™ FX; BD Diagnostics Systems, Sparks, MD). The BCBs were processed and tested according to the manufacturer's instructions. As our laboratory operates an integrated 24-hour active service, positive blood cultures are routinely assessed and reported by a clinical microbiologist. When bottles were flagged as being positive, they were removed from the data units and analyzed using direct MALDI-TOF MS and routine MALDI-TOF MS.

## Routine MALDI-TOF MS identification

When a positive BCB was detected, one drop of sample from the positive BCB was taken for subculture on BAP agar at 35˚C with 5% $CO_2$. Once bacterial isolate colonies had grown on the BAP agar, a single microbial colony from an agar plate was smeared on a MALDI steel target plate (Bruker Daltonics, Bremen, Germany) using a sterile toothpick, and then overlaid with 1 μl of 70% (v/v) aqueous formic acid followed by air-drying. This was then overlaid with 1 μl of matrix (α-cyano-4-hydroxycinnamic acid (HCCA) in 50% (v/v) aqueous acetonitrile containing 2.5% (v/v) trifluoroacetic acid (TFA)) (HCCA; Bruker Daltonics, Bremen, Germany), dried in air, and then loaded into the mass spectrometer for identification.

## Processing methods for direct MALDI-TOF MS identification

For direct MALDI-TOF MS-based identification of the pathogens, positive blood cultures were sampled. A Serum Separator Tube (BD Diagnostics) was inoculated with 5 ml of broth from a positive BCB. The tube was then centrifuged at 3000 rpm for 15 minutes to sediment blood cells under the separator gel. After the supernatant had been discarded, the bacterial pellet remaining above the separator gel was resuspended in 300 μl of sterile distilled water, mixed slowly, and transferred to a new 1.5-ml polypropylene tube (Eppendorf, Hamburg, Germany). The tube was centrifuged again at 3000 rpm for 10 minutes to pellet the bacterial cells. The supernatant was then discarded, and 300 μl of deionized water and 900 μl of ethanol were added to resuspend the pellet. The suspension obtained following this sample preparation method was centrifuged at 13000 rpm for 2 minutes, and the supernatant was discarded. The pellet was then centrifuged at 13000 rpm for another 2 minutes to remove residual ethanol, followed by the addition of 50 μl of formic acid (70% v/v) and 50 μl of 100% acetonitrile, with the mixture being mixed thoroughly after each reagent had been added. The suspension was centrifuged again at 13000 rpm for another 2 minutes, and 1 μl of the supernatant was spotted onto the MALDI steel target plate. MALDI-TOF MS analysis was performed following air-drying of 1 μl HCCA matrix solution placed onto the dried sample spot in duplicate [21].

## MALDI-TOF MS analysis

Mass spectra profiles were acquired using a Microflex LT MALDI-TOF mass spectrometer system (Bruker Daltonics, Bremen, Germany) following the manufacturer's settings. Spectra were recorded in the linear positive mode at a laser frequency of 60 Hz within a mass ranging from 2000 Da to 20000 Da. All bacteria identification was performed using Bruker MALDI Biotyper 3.1 software and library (5989 isolates; Bruker Daltonics). The criteria used for microorganism analysis and identification were as recommended by the manufacturer. Each measurement

was performed only once for each specimen. The time to identification of the bacterial strain was determined as the time from colony formation to the time at which the results were reported to a physician. Identification was considered as being at the species level with a score of > 2.0; if the score was between 1.7 and 2.0, identification was regarded as being at genus-level, while a score below 1.7 was considered as being unreliable identification.

## Statistical analysis

Statistical analysis was performed using MedCalc software (version 17.9.2; MedCalc, Ostend, Belgium). As a preliminary Kolmogorov-Smirnov test demonstrated that the samples did not follow a normal distribution, we decided to use the nonparametric Kruskal-Wallis test to compare the time to microorganism identification between different diagnostic methods. In order to fully understand group differences, the Bonferroni post hoc test was used for multiple comparisons when the Kruskal-Wallis test was significant. All results are expressed as medians and interquartile ranges (IQRs). Post hoc power analysis showed that a sample size of 80 was sufficient to achieve 89% power to detect an effect size of 0.23 with a significance level of 0.05 using a two-sided one-way ANOVA test (G Power v3.1, Baden-Württemberg, Germany). The level of statistical significance was set at $p < 0.05$.

## Results

### Patient characteristics

Two patients had recurrent infections after a two-stage exchange protocol, and one patient had bilateral staged knee PJIs at different time points during the study period. Therefore, 80 SF samples were taken and analyzed from 77 patients who fulfilled the MSIS criteria (Table 1).

**Table 1. Characteristics of the 77 patients (80 joints) analyzed.**

| | Patients who met the MSIS criteria | |
|---|---|---|
| Median age, years (IQR) | 67 | (57–77.8) |
| Sex, n (%) | | |
| Male | 36 | (47) |
| Female | 41 | (53) |
| Median BMI, kg/m$^2$ (IQR) | 25.4 | (21.6–29.5) |
| Joint, n (%) | | |
| Hip | 41 | (51) |
| Knee | 39 | (49) |
| Prosthesis, n (%) | | |
| Primary | 60 | (75) |
| Revision | 20 | (25) |
| Median ASA (IQR) | 3 | (2–3) |
| PJI type, n (%) | | |
| Acute | 11 | (14) |
| Chronic | 69 | (86) |
| Median serum ESR, mm/h (IQR) | 60 | (34–90) |
| Median serum CRP, mg/dL (IQR) | 47 | (10.3–113.8) |
| Median synovial WBC count, cells/μL (IQR) | 8280 | (940–37180) |
| Median synovial PMN (%), (IQR) | 87 | (76–93) |

MSIS, Musculoskeletal Infection Society; BMI, body mass index; ASA, American Society of Anesthesiologists; PJI, prosthetic joint infection; ESR, erythrocyte sedimentation rate; CRP, C-reactive protein; WBC, white blood cell; PMN, polymorphonuclear neutrophil; IQR, interquartile range.

The median age of the patients was 67.0 years (IQR: 57.0–77.8 years) and the median body mass index was 25.4 kg/m$^2$ (IQR: 21.6–29.5 kg/m$^2$). Forty-one patients (53%) were female. The PJI cases included 41 hip arthroplasties (51%) and 39 knee arthroplasties (49%). Overall, 75% of the patients had a primary prothesis, 86% had a chronic infection and 14% had an acute infection. The median erythrocyte sedimentation rate was 60 mm/h (IQR: 34–90 mm/h) and the median C-reactive protein level was 47 mg/dL (IQR: 10.3–113.8 mg/dL). The mean synovial WBC count was 8280 cells/μL (IQR: 940–37180 cells/μL) and the synovial PMN % was 87% (IQR: 76–93%).

## Direct MALDI-TOF MS versus routine MALDI-TOF MS

Table 2 shows the results of diagnostic testing using the three methods. Of the 80 PJIs, direct and routine MALDI-TOF MS both detected 64 positive cultures (80%), while conventional swabs only identified 47 positive cultures (59%, p = 0.002). Direct MALDI-TOF MS identified 85.3% of the gram-positive organisms, of which *Staphylococcus aureus* (n = 22) and *Staphylococcus epidermidis* (n = 8) accounted for the majority. In addition, direct MALDI-TOF MS identified 92.3% of the gram-negative organisms. No fungi were identified by direct MALDI-TOF MS. Of the 80 samples identified by direct MALDI-TOF MS, 63 (79%) had a result concordant with the routine MALDI-TOF MS results. The median score was 2.31 (IQR: 2.12–2.39) for

**Table 2. Microorganisms identified by diagnostic testing of synovial fluid in positive blood culture bottles using direct MALDI-TOF MS, routine MALDI-TOF MS, and conventional swabs.**

| Organisms | Direct MALDI-TOF MS (%) | Routine MALDI-TOF MS (%) | Swab (%) |
|---|---|---|---|
| Culture-positive | 64 (80) | 64 (80) | 47 (59) |
| Microorganisms | | | |
| Gram-positive | 35 (44) | 41 (51) | 32 (40) |
| *Staphylococcus aureus* | 22 (28) | 22 (28) | 21 (26) |
| *Staphylococcus epidermidis* | 8 (10) | 10 (13) | 5 (6) |
| Coagulase-negative staphylococci | 1 (1) | 4 (5) | 2 (3) |
| *Staphylococcus lugdunensis* | 2 (3) | 2 (3) | 1 (1) |
| *Enterococcus faecalis* | 1 (1) | 1 (1) | 1 (1) |
| *Streptococcus anginosus* | 1 (1) | 1 (1) | 1 (1) |
| *Cutibacterium acnes* | 0 | 1 (1) | 0 |
| *Streptococcus oralis* | 0 | 0 | 1 (1) |
| Gram-negative | 12 (15) | 13 (16) | 9 (11) |
| *Escherichia coli* | 4 (5) | 4 (5) | 3 (4) |
| *Klebsiella pneumoniae* | 4 (5) | 4 (5) | 2 (3) |
| *Pseudomonas aeruginosa* | 3 (4) | 3 (5) | 2 (3) |
| *Enterobacter cloacae* | 1 (1) | 1 (1) | 1 (1) |
| *Bacteroides fragilis* | 0 | 0 | 1 (1) |
| *Sphingobacterium thalpophilum* | 0 | 1 (1) | 0 |
| Fungus | 0 | 2 (3) | 3 (4) |
| *Candida parapsilosis* | 0 | 2 (3) | 2 (3) |
| *Candida glabrata* | 0 | 0 | 1 (1) |
| Polymicrobial | 0 | 8 (10) | 3 (4) |
| Failed identification | 17 (21) | 0 | 0 |
| Culture-negative | 16 (20) | 16 (20) | 33 (41) |

MALDI-TOF MS, matrix-assisted laser desorption ionization time-of-flight mass spectrometry.

**Table 3. Discordant identification of organisms between each method.**

| Direct MALDI-TOF MS | Routine MALDI-TOF MS | Conventional swabs |
|---|---|---|
| Failed identification | *Staphylococcus epidermidis* | No growth |
| Failed identification | *Staphylococcus epidermidis* | No growth |
| Failed identification | Coagulase-negative staphylococci | No growth |
| Failed identification | Coagulase-negative staphylococci | Coagulase-negative staphylococci |
| Failed identification | Coagulase-negative staphylococci | Polymicrobial (*Staphylococcus epidermidis*, *Klebsiella pneumoniae*) |
| Failed identification | *Cutibacterium acnes* | No growth |
| Failed identification | *Sphingobacterium thalpophilum* | No growth |
| Failed identification | *Candida parapsilosis* | *Candida parapsilosis* |
| Failed identification | *Candida parapsilosis* | *Candida parapsilosis* |
| Failed identification | Polymicrobial (*Bacteroides fragilis*, Coagulase-negative staphylococci) | *Bacteroides fragilis* |
| Failed identification | Polymicrobial (*Enterococcus faecalis*, Coagulase-negative staphylococci) | No growth |
| Failed identification | Polymicrobial (*Candida glabrata*, Coagulase-negative staphylococci) | *Candida glabrata* |
| Failed identification | Polymicrobial (*Staphylococcus epidermidis*, *Corynebacterium minutissimum*) | No growth |
| Failed identification | Polymicrobial (*Staphylococcus aureus*, *Enterococcus faecalis*) | *Staphylococcus aureus* |
| Failed identification | Polymicrobial (*Streptococcus oralis*, *Staphylococcus epidermidis*) | *Streptococcus oralis* |
| Failed identification | Polymicrobial (*Staphylococcus aureus*, *Staphylococcus epidermidis*) | *Staphylococcus aureus* |
| Failed identification | Polymicrobial (*Enterococcus faecalis*, *Klebsiella pneumoniae*, *Morganella morganii*) | Polymicrobial (*Enterococcus faecalis*, *Enterobacter cloacae*, *Morganella morganii*) |
| *Staphylococcus aureus* | *Staphylococcus aureus* | No growth |
| *Staphylococcus aureus* | *Staphylococcus aureus* | No growth |
| *Staphylococcus aureus* | *Staphylococcus aureus* | No growth |
| *Staphylococcus epidermidis* | *Staphylococcus epidermidis* | No growth |
| *Staphylococcus epidermidis* | *Staphylococcus epidermidis* | No growth |
| Coagulase-negative staphylococci | Coagulase-negative staphylococci | No growth |
| *Staphylococcus lugdunensis* | *Staphylococcus lugdunensis* | No growth |
| *Escherichia coli* | *Escherichia coli* | No growth |
| *Klebsiella pneumoniae* | *Klebsiella pneumoniae* | No growth |
| *Klebsiella pneumoniae* | *Klebsiella pneumoniae* | No growth |
| *Pseudomonas aeruginosa* | *Pseudomonas aeruginosa* | No growth |
| *Staphylococcus epidermidis* | *Staphylococcus epidermidis* | Polymicrobial (*Pseudomonas aeruginosa*, *Enterobacter cloacae*) |
| *Klebsiella pneumoniae* | *Klebsiella pneumoniae* | *Staphylococcus aureus* |
| *Staphylococcus aureus* | *Staphylococcus aureus* | Coagulase-negative staphylococci |
| No growth | No growth | *Klebsiella pneumoniae* |

MALDI-TOF MS, matrix-assisted laser desorption ionization time-of-flight mass spectrometry.

direct MALDI-TOF MS identification. Among the samples, only two cases of *Staphylococcus epidermidis* were identified at the genus level, with a score of < 2. Seventeen samples were flagged as being positive in direct MALDI-TOF MS but identification failed, whereas they were all identified with routine MALDI-TOF MS (Table 3); these cases included two *Staphylococcus epidermidis*, three coagulase-negative staphylococci, one *Cutibacterium acnes*, one *Sphingobacterium thalpophilum*, two fungi, and eight polymicrobial organisms.

### Direct MALDI-TOF MS versus conventional swab culture

Of the 80 samples that were identified using direct MALDI-TOF MS, 48 (60%) had a result concordant with conventional swab cultures and 32 had discordant results (Table 3). Among these samples, 17 were flagged as being positive but identification failed with direct MALDI-TOF MS, and 11 samples showed negative conventional swab culture results but cultures were detected using direct MALDI-TOF MS, including three *Staphylococcus aureus*, two *Staphylococcus epidermidis*, one coagulase-negative staphylococci, one *Staphylococcus lugdunensis*, one *Escherichia coli*, two *Klebsiella pneumoniae* and one *Pseudomonas aeruginosa*. Three PJI cases grew polymicrobial organisms, of which conventional swab cultures identified *Staphylococcus aureus* and coagulase-negative staphylococci, but direct MALDI-TOF MS identified *Staphylococcus epidermidis*, *Klebsiella pneumoniae* and *Staphylococcus aureus*.

### Routine MALDI-TOF MS versus conventional swab culture

The most commonly identified organism was *Staphylococcus aureus* in the three groups. Eighteen samples analyzed using BCBs had microbial growth where traditional swab cultures showed none. Of these 18 samples, the most significant finding was the successful identification of *Staphylococcus epidermidis* in four samples, followed by three *Staphylococcus aureus*, two coagulase-negative staphylococci, two *Klebsiella pneumoniae*, two polymicrobial organisms, one *Cutibacterium acnes*, one *Staphylococcus lugdunensis*, one *Escherichia coli*, one *Pseudomonas aeruginosa* and one *Sphingobacterium thalpophilum* (Table 3).

### Time to identification

The median time to identification of a bacterial strain was 12.7 hours (IQR: 8.9–19.6 hours) for indirect MALDI-TOF MS, 39.5 hours (IQR: 22.8–46.0 hours) for routine MALDI-TOF MS, and 44.4 hours (IQR: 27.2–71.6 hours) for swab cultures. The overall difference between the three groups was significant ($p < 0.0001$). Significant Bonferroni-adjusted post hoc interactions are shown in Fig 2. In pairwise comparisons, there were significant differences in the time of microorganism identification between direct MALDI-TOF MS and routine MALDI-TOF MS ($p < 0.0001$) or swab culture ($p < 0.0001$). There was no significant difference between routine MALDI-TOF MS and swab culture ($p = 0.0268$).

## Discussion

In this study, we found that direct MALDI-TOF MS identified a similar rate of microorganisms from positive BCBs of cultured SF from patients with PJIs as conventional swabs. However, direct MALDI-TOF MS had a lower pathogen identification rate than routine MALDI-TOF MS. Direct MALDI-TOF MS also significantly shortened the time to strain identification compared to routine MALDI-TOF MS and conventional swabs, suggesting that physicians could obtain strain reports sooner using direct MALDI-TOF MS after surgery to allow for prompt and appropriate antibiotic treatment and more favorable patient outcomes.

Direct MALDI-TOF MS identified 85.3% of the gram-positive organisms and 92.3% of the gram-negative organisms in SF samples in BCBs. Our results are consistent with previous studies which showed that direct MALDI-TOF MS had high accuracy for gram-negative bacteria but moderate accuracy for gram-positive organisms [17, 22, 23]. Zhou et al. demonstrated a higher accuracy for gram-negative bacteria than for gram-positive bacteria (92.8 vs. 82.4%) in their in-house protocol for direct MALDI-TOF MS of positive BCBs [8]. In addition, French et al reported an identification rate of 53% for gram-positive organisms and 89% for gram-negative organisms [12]. A recent systemic review and meta-analysis of 32 studies reported an

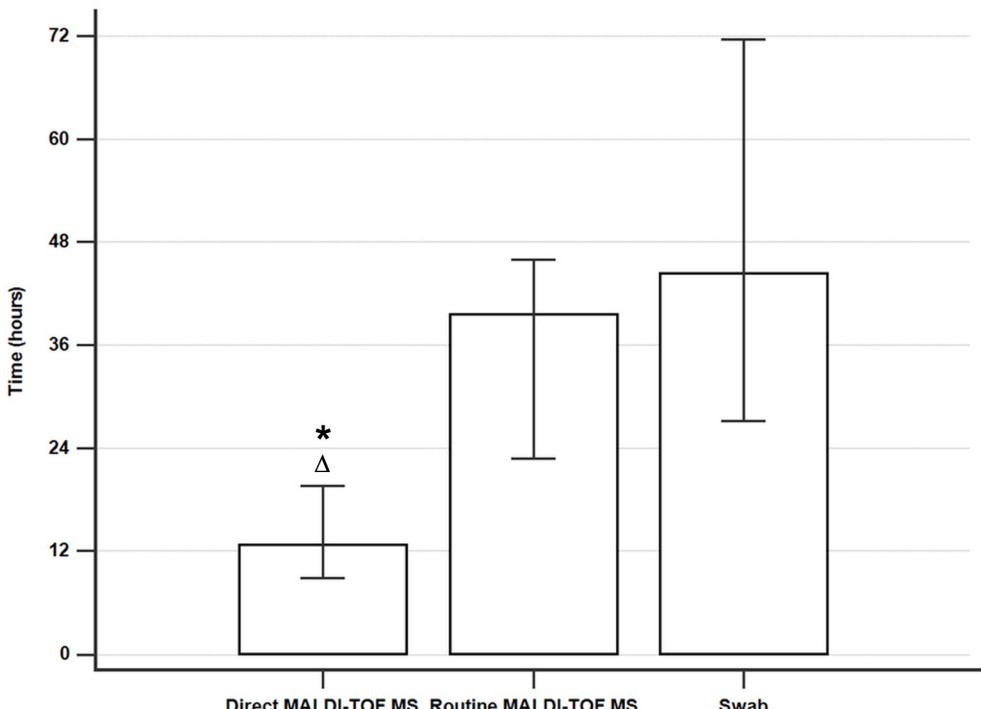

**Fig 2. Median time to identification of microorganisms.** The error bars represent the 25th and 75th percentiles. *
Significant Bonferroni-adjusted pairwise comparisons compared to the routine MALDI-TOF MS. Δ Significant
Bonferroni-adjusted pairwise comparisons compared to the swabs.

overall correct identification rate of 72% for gram-positive bacteria compared to 92% for
gram-negative bacteria with direct bacterial identification from positive BCBs using MALDI-
TOF MS [24]. A possible reason for the lower identification rate of gram-positive bacteria with
MALDI-TOF MS analysis is because gram-positive bacteria require pre-processing steps due
to their more robust cell wall, which decreases the protein extraction rate [25].

Cultures of SF in BCBs have been used for bacterial identification in PJIs, and 3–5 days are
usually required to identify the final pathogen [4, 5]. If the microorganism can be identified
earlier, targeted empirical treatment can be administered more precisely to reduce the emer-
gence of resistant strains. Direct MALDI-TOF MS analysis of positive blood cultures has the
advantage of rapid pathogen identification. One study showed that the median time to identifi-
cation by direct MALDI-TOF MS of positive blood cultures was significantly shorter (7.1
hours) than by standard methods (48.1 hours) [26]. In the current study, direct MALDI-TOF
MS pathogen identification in SF specimens in BCBs reduced the pathogen identification time
to a median duration of 12.7 hours, which was significantly lower than the median duration of
39.5 hours for routine MALDI-TOF MS and 44.4 hours for conventional swabs. Therefore,
direct MALDI-TOF MS saved approximately 27 hours and 32 hours in the process of bacterial
identification compared with routine MALDI-TOF MS and conventional swabs, respectively,
in patients with PJIs.

Interestingly, three candida species were identified in conventional swab cultures, however
none of these fungi were identified using direct MALDI-TOF MS after the BCBs had been
flagged positive. We speculate that inadequate fungal protein was present in the SF samples
after incubation in BCBs [27], and that subculture resulted in an increase in fungal protein
which was then sufficient for identification. Routine MALDI-TOF MS identified two cases of

*Candida parapsilosis* and one of *Candida glabrata* in the polymicrobial infections in these three cases.

Identification failed in 17 SF samples (21%) with direct MALDI-TOF MS, whereas these samples were successfully identified with routine MALDI-TOF MS. Of these samples, almost 50% were polymicrobial infections. In addition, eight samples were identified as containing polymicrobial organisms by routine MALDI-TOF MS, however direct MALDI-TOF MS did not identify the pathogens in any of these eight samples. When polymicrobial samples are subjected to MALDI-TOF MS identification, an aberrant protein spectrum is produced, which results in a mixture of several profiles. In such cases, MALDI-TOF MS cannot identify the microorganisms from its database.

The strength of this study is the strict prospective design and that all participants fulfilled the ICM criteria for the diagnosis of PJIs. The weakness of the study is that it did not include non-infected patients as a control group; therefore, we were unable to provide the specificity of each method. However, a previous study showed that SF cultured in BCBs had higher sensitivity and specificity compared with swab samples for the diagnosis of PJIs [4]. Moreover, the aim of this study was to focus on the rapid identification of PJI organisms through different methods using SF samples in BCBs. Further research is required to examine the sensitivity and specificity of these diagnostic methods. Another weakness may have resulted from antimicrobial selection, as we did not record changes in antimicrobial agents before and after microorganism identification. However, an infectious disease specialist was consulted when the culture report was available. According to other studies, more narrow-spectrum antibiotic therapy can be administered on day one, even without susceptibility testing, once the microorganisms have been identified by direct MALDI-TOF MS [12].

## Conclusion

Direct MALDI-TOF MS can be used to rapidly identify a single microorganism in patients with PJI following the culture of SF in BCBs. For fungal or polymicrobial PJIs, routine MALDI-TOF MS is a better diagnostic tool than direct MALDI-TOF MS. Further large-scale studies are required to verify the clinical impact and the value of this testing method.

## Supporting information

**S1 Checklist. CONSORT 2010 checklist of information to include when reporting a randomised trial**∗**.**
(DOC)

**S1 File.**
(DOC)

**S1 Dataset.**
(XLS)

## Acknowledgments

We acknowledge and thank the Biostatistics Center at Kaohsiung Chang Gung Memorial Hospital for their statistical work.

## Author Contributions

**Conceptualization:** Feng-Chih Kuo, Chen-Hsiang Lee.

**Data curation:** Feng-Chih Kuo, Chun-Chih Chien, Mel S. Lee, Jun-Wen Wang, Po-Chun Lin.

**Investigation:** Feng-Chih Kuo, Mel S. Lee, Po-Chun Lin, Chen-Hsiang Lee.

**Methodology:** Feng-Chih Kuo, Chun-Chih Chien, Mel S. Lee, Chen-Hsiang Lee.

**Resources:** Mel S. Lee, Jun-Wen Wang, Po-Chun Lin.

**Software:** Chun-Chih Chien.

**Validation:** Chun-Chih Chien, Jun-Wen Wang, Chen-Hsiang Lee.

**Visualization:** Feng-Chih Kuo, Jun-Wen Wang.

**Writing – original draft:** Feng-Chih Kuo, Mel S. Lee, Po-Chun Lin.

**Writing – review & editing:** Jun-Wen Wang, Chen-Hsiang Lee.

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
