## [Decision Letter · Decision Letter 0]

25 Jun 2020

PONE-D-20-14441

Rapid diagnosis of periprosthetic joint infection from synovial fluid in blood culture bottles by direct matrix-assisted laser desorption ionization time-of-flight mass spectrometry

PLOS ONE

Dear Dr. Lee,

Thank you for submitting your manuscript to PLOS ONE. After careful consideration, we feel that it has merit but does not fully meet PLOS ONE’s publication criteria as it currently stands. Therefore, we invite you to submit a revised version of the manuscript that addresses the points raised during the review process.

The manuscript needs to be improved resolving the criticisms reported by the Authors. In particular the study design should be better clarified.

We look forward to receiving your revised manuscript.

Kind regards,

Adriana Calderaro

Academic Editor

PLOS ONE

Journal Requirements:

a) Did participants provide their written or verbal informed consent to participate in this study?

3. Please update the title page within your main document to list all authors and all affiliations as per our author instructions and clearly indicate the corresponding author.

Reviewers' comments:

Reviewer's Responses to Questions

**Comments to the Author**

1. Is the manuscript technically sound, and do the data support the conclusions?

Reviewer #1: Yes

Reviewer #2: Partly

2. Has the statistical analysis been performed appropriately and rigorously? 

Reviewer #1: Yes

Reviewer #2: Yes

3. Have the authors made all data underlying the findings in their manuscript fully available?

Reviewer #1: Yes

Reviewer #2: Yes

4. Is the manuscript presented in an intelligible fashion and written in standard English?

Reviewer #1: Yes

Reviewer #2: No

5. Review Comments to the Author

Reviewer #1: The authors collect data from 77 patients to evaluate a direct matrix-assisted laser desorption ionization time-of-flight mass spectrometry (MALDI-TOF MS) in compared with the current laboratory practices. The results showed that the new technique shortens the time to microorganism identification and has superior results. The manuscript was well prepared.

1. Line 143. “80 SF samples were taken from 77 patients…” In other words, some samples were taken from the same patients and were correlated. However, the analysis doesn’t discuss how the correlated data were accounted.

2. Line 195. “All pairwise were significant differences …..”. Please report ANOVA results first. Also, what multiple testing procedure was applied?

Reviewer #2: The paper entitled "Rapid diagnosis of periprosthetic joint infection from synovial fluid in blood culture bottles by direct matrix-assisted laser desorption ionization time-of-flight mass spectrometry" aimed to investigate the clinical practice for integrating a direct MALDI-TOF MS identification for rapid diagnosis of PJI. The study was a strict prospective design, but there are several major limitations.

1: The numbers of the patients included in the study is not big enough, there is still needed for the more samples to further verify the conclusion.

2: The presentation of the study is unclear and lacks rigor.

3: The authors argue that the purpose of the study was to evaluate whether direct MALDI-TOF MS can shorten the time to preliminary strain identification as compare with routine methods (Page 4, Line 61-62). Direct MALDI-TOF MS of course can shorten the time to preliminary strain identification, as subculture is not needed. Thus the argument is meaningless.

4: The Material and Methods section has several mistakes and detailed information is lacked. Page 6, line 116, where is the ethanol come from? Line 116-120, to my knowledge, it should be “μL” instead of “mL”, etc…..

5: The results section is badly structure and the description is confusing, major improvement is needed.

6. PLOS authors have the option to publish the peer review history of their article (what does this mean?). If published, this will include your full peer review and any attached files.

Reviewer #1: No

Reviewer #2: No

---

## [Author Response · Author response to Decision Letter 0]

23 Jul 2020

Response to Reviewers

Re: Rapid diagnosis of periprosthetic joint infection from synovial fluid in blood culture bottles by direct matrix-assisted laser desorption ionization time-of-flight mass spectrometry

Dear Editors,

Thank you for the comprehensive review of our manuscript. We appreciate the opportunity to submit a revised version and found the reviewers' comments to be very insightful and contribute considerably to the improvement of our manuscript.

Below we provide our point-by-point responses to all of the reviewers’ comments. We also send this manuscript for English editing. We hope you will find our revised manuscript suitable for publication in PLOS ONE.

Sincerely,

Chen-Hsiang Lee

Reviewer#1

1. Line 143. “80 SF samples were taken from 77 patients...” In other words, some samples were taken from the same patients and were correlated. However, the analysis doesn’t discuss how the correlated data were accounted.

Ans: Thank you for this and all of your comments. We have rewritten this in page 13, line 158-159: “Two patients had recurrent infections after a two-stage exchange protocol. One patient had bilateral staged knee PJIs at different time points during the study period. Therefore, 80 SF samples were taken and analyzed from 77 patients who fulfilled the MSIS criteria (Table 1).”

2. Line 195. “All pairwise were significant differences .....”. Please report ANOVA results first. Also, what multiple testing procedure was applied?

Ans: We used the Bonferroni post hoc test for multiple comparisons between groups. We have added this to page 12, line 149-151: “In order to fully understand group differences, the Bonferroni post hoc test was used for multiple comparisons when the Kruskal-Wallis test was significant.”

We reported the overall results and group differences on page 20, line 211-226: “The overall difference between the three groups was significant (p < 0.0001). Significant Bonferroni-adjusted post hoc interactions are shown in Fig 2. In pairwise comparisons, there were significant differences in the time of microorganism identification between direct MALDI-TOF MS and routine MALDI-TOF MS (p < 0.0001) or swab culture (p < 0.0001). There was no significant difference between routine MALDI-TOF MS and swab culture (p = 0.0268).”

Reviewer #2

1. The numbers of the patients included in the study is not big enough, there is still needed for the more samples to further verify the conclusion.

Ans: Thank you for this and all of your comments. We have added this to address the power analysis on page 12, line 152-154: “Post hoc power analysis showed that a sample size of 80 was sufficient to achieve 89% power to detect an effect size of 0.23 with a significance level of 0.05 using a 2-sided one-way ANOVA test (G Power v3.1, Baden-Württemberg, Germany).”

We have also added this in the conclusion on page 26, line 287-288: “Further large-scale studies are required to verify the clinical impact and the value of this testing method.”

2. The presentation of the study is unclear and lacks rigor.

Ans: We calculated group differences using the Bonferroni post hoc test for multiple comparisons (page 12, line 149-151): “In order to fully understand group differences, the Bonferroni post hoc test was used for multiple comparisons when the Kruskal-Wallis test was significant.”

We reported the overall results and group differences on page 20, line 211-216: “The overall difference between the three groups was significant (p < 0.0001). Significant Bonferroni-adjusted post hoc interactions are shown in Fig 2. In pairwise comparisons, there were significant differences in the time of microorganism identification between direct MALDI-TOF MS and routine MALDI-TOF MS (p < 0.0001) or swab culture (p < 0.0001). There was no significant difference between routine MALDI-TOF MS and swab culture (p = 0.0268).”

3. The authors argue that the purpose of the study was to evaluate whether direct MALDI-TOF MS can shorten the time to preliminary strain identification as compare with routine methods (Page 4, Line 61-62). Direct MALDI-TOF MS of course can shorten the time to preliminary strain identification, as subculture is not needed. Thus the argument is meaningless.

Ans: Thank you for your valuable opinion. We have rewritten this part of the introduction on page 6, line 52-59: “Theoretically, direct MALDI-TOF MS can decrease the time to preliminary strain identification, as a subculture is not required in this process. This rapid microbial identification method has been used in many infection fields, such as bloodstream infections [8,11,12] meningitis [13,14] and urinary traction infections [15,16]. This method has been shown to have high accuracy for Gram-negative organisms but relatively low accuracy for Gram-positive bacteria [17]. However, Gram-positive bacteria are the most common organisms that cause PJIs [18,19]. However, whether the direct identification method can be efficiently applied in patients with PJIs by culturing SF in BCBs for a rapid and accurate diagnosis has yet to be elucidated.”

4. The Material and Methods section has several mistakes and detailed information is lacked. Page 6, line 116, where is the ethanol come from? 

Ans: We have rewritten the section of Material and Methods (line 89 on page 8 to line 143 on page 11) and changed a reference [Yonetani S, Ohnishi H, Ohkusu K, Matsumoto T, Watanabe T. Direct identification of microorganisms from positive blood cultures by MALDI-TOF MS using an in-house saponin method. Int J Infect Dis. 2016;52: 37–42. doi:10.1016/j.ijid.2016.09.014].

5. Line 116-120, to my knowledge, it should be “μL” instead of “mL”, etc.....

Ans: Thank you. The unit has been changed to μL. 

6. The results section is badly structure and the description is confusing, major improvement is needed.

Ans: The results section describes direct versus routine MALDI-TOF MS, followed by direct MALDI-TOF MS versus swabs, then routine swab culture versus swab.

We have also added the group difference in the time to microorganism identification on page 20, line 211-216: “The overall difference between the three groups was significant (p < 0.0001). Significant Bonferroni-adjusted post hoc interactions are shown in Fig 2. In pairwise comparisons, there were significant differences in the time of microorganism identification between direct MALDI-TOF MS and routine MALDI-TOF MS (p < 0.0001) or swab culture (p < 0.0001). There was no significant difference between routine MALDI-TOF MS and swab culture (p = 0.0268).”

We also added a paragraph in the discussion section to describe our finding in page 22 and 23, line 232-244 “Interestingly, direct MALDI-TOF MS identified 85.3% of Gram-positive organisms and 92.3% of Gram-negative organisms in SF samples in BCBs. Our results are consistent with previous studies which showed that direct MALDI-TOF MS had high accuracy for Gram-negative bacteria but moderate accuracy for Gram-positive organisms [17,22,23]. Zhou et al. demonstrated a higher accuracy for Gram-negative bacteria than for Gram-positive bacteria (92.8 vs. 82.4%) in their in-house protocol for direct MALDI-TOF MS in positive BCBs [8]. In addition, French et al reported an identification rate of 53% for Gram-positive organisms and 89% for Gram-negative organisms [12]. A recent systemic review and meta-analysis of 32 studies reported an overall correct identification rate of 72% for Gram-positive bacteria compared to 92% for Gram-negative bacteria with direct bacterial identification from positive BCBs using MALDI-TOF MS [24]. A possible reason for the lower identification rate of Gram-positive bacteria with MALDI-TOF MS analysis is because Gram-positive bacteria require pre-processing steps due to their more robust cell wall, which decreases the protein extraction rate [25].”

---

## [Decision Letter · Decision Letter 1]

6 Aug 2020

PONE-D-20-14441R1

Rapid diagnosis of periprosthetic joint infection from synovial fluid in blood culture bottles by direct matrix-assisted laser desorption ionization time-of-flight mass spectrometry

PLOS ONE

Dear Dr. Lee,

Thank you for submitting your manuscript to PLOS ONE. After careful consideration, we feel that it has merit but does not fully meet PLOS ONE’s publication criteria as it currently stands. Therefore, we invite you to submit a revised version of the manuscript that addresses the points raised during the review process.

The manuscript was improved, however it requires again a revision. In particular, a language and editing revsion is needed especially for the sentences newly added. Furthermore, the Authors should be more cautious affirming that a subculturing is not needed when the direct identification by MALDI-TOF MS is performed.

We look forward to receiving your revised manuscript.

Kind regards,

Adriana Calderaro

Academic Editor

PLOS ONE

Reviewers' comments:

Reviewer's Responses to Questions

**Comments to the Author**

1. If the authors have adequately addressed your comments raised in a previous round of review and you feel that this manuscript is now acceptable for publication, you may indicate that here to bypass the “Comments to the Author” section, enter your conflict of interest statement in the “Confidential to Editor” section, and submit your "Accept" recommendation.

Reviewer #1: All comments have been addressed

2. Is the manuscript technically sound, and do the data support the conclusions?

Reviewer #1: (No Response)

3. Has the statistical analysis been performed appropriately and rigorously? 

Reviewer #1: (No Response)

4. Have the authors made all data underlying the findings in their manuscript fully available?

Reviewer #1: (No Response)

5. Is the manuscript presented in an intelligible fashion and written in standard English?

Reviewer #1: (No Response)

6. Review Comments to the Author

Reviewer #1: (No Response)

7. PLOS authors have the option to publish the peer review history of their article (what does this mean?). If published, this will include your full peer review and any attached files.

Reviewer #1: No

---

## [Author Response · Author response to Decision Letter 1]

17 Aug 2020

Editor comment

The manuscript was improved, however it requires again a revision. In particular, a language and editing revsion is needed especially for the sentences newly added. Furthermore, the Authors should be more cautious affirming that a subculturing is not needed when the direct identification by MALDI-TOF MS is performed.

Ans: Thank you for this and all of your comments. The revised edition has been edited by an English native speaker to improve the language writing.

Thanks for your reminder. The most time-intensive process using routine MALDI-TOF MS is the time required for the subculture of specimens before identification. Theoretically, direct MALDI-TOF MS can decrease the time to preliminary strain identification. We have clarified the issue of performing a subculture in the revised version of the manuscript.

---

## [Decision Letter · Decision Letter 2]

3 Sep 2020

Rapid diagnosis of periprosthetic joint infection from synovial fluid in blood culture bottles by direct matrix-assisted laser desorption ionization time-of-flight mass spectrometry

PONE-D-20-14441R2

Dear Dr. Lee,

We’re pleased to inform you that your manuscript has been judged scientifically suitable for publication and will be formally accepted for publication once it meets all outstanding technical requirements.

Kind regards,

Adriana Calderaro

Academic Editor

PLOS ONE

Additional Editor Comments (optional):

Reviewers' comments:

Reviewer's Responses to Questions

**Comments to the Author**

1. If the authors have adequately addressed your comments raised in a previous round of review and you feel that this manuscript is now acceptable for publication, you may indicate that here to bypass the “Comments to the Author” section, enter your conflict of interest statement in the “Confidential to Editor” section, and submit your "Accept" recommendation.

Reviewer #1: All comments have been addressed

2. Is the manuscript technically sound, and do the data support the conclusions?

Reviewer #1: (No Response)

3. Has the statistical analysis been performed appropriately and rigorously? 

Reviewer #1: (No Response)

4. Have the authors made all data underlying the findings in their manuscript fully available?

Reviewer #1: (No Response)

5. Is the manuscript presented in an intelligible fashion and written in standard English?

Reviewer #1: (No Response)

6. Review Comments to the Author

Reviewer #1: (No Response)

7. PLOS authors have the option to publish the peer review history of their article (what does this mean?). If published, this will include your full peer review and any attached files.

Reviewer #1: No

---

## [Editor Report · Acceptance letter]

11 Sep 2020

PONE-D-20-14441R2 

Rapid diagnosis of periprosthetic joint infection from synovial fluid in blood culture bottles by direct matrix-assisted laser desorption ionization time-of-flight mass spectrometry 

Dear Dr. Lee:

I'm pleased to inform you that your manuscript has been deemed suitable for publication in PLOS ONE. Congratulations! Your manuscript is now with our production department. 

Kind regards, 

on behalf of

MD, PhD, Associate Professor Adriana Calderaro 

Academic Editor

PLOS ONE